# Waste of Fresh Fruits in Yaoundé, Cameroon: Challenges for Retailers and Impacts on Consumer Health

**Aristide Guillaume Kamda Silapeux** [1,2,3,*], **Roger Ponka** [4], **Chiara Frazzoli** [5] and **Elie Fokou** [2,3]

1 Department of Social Economy and Family Management, HTTTC, University of Buea, P.O. Box 249, Kumba, Cameroon

2 Nutrition, Food Safety and Wholesomeness. Prevention, Education and Research Network, P.O. Box 3743 Yaoundé, Cameroon

3 Laboratory for Food Sciences and Metabolism, Department of Biochemistry, University of Yaoundé I, P.O. Box 812 Yaoundé, Cameroon; efokou@yahoo.com

4 Department of Agriculture, Livestock and Derivated Products, National Advanced School of Engineering of Maroua, University of Maroua, P.O. Box 46 Maroua, Cameroon; rponka@yahoo.fr

5 Department of Cardiovascular and endocrine-metabolic diseases, and ageing, Istituto Superiore di Sanità, Viale Regina Elena 299, 00161 Rome, Italy; chiara.frazzoli@iss.it

\* Correspondence: aristide.kamda@noodlesonlus.org; Tel.: +237-652053684

**Abstract:** Post-harvest losses contribute significantly to food insecurity and affect the nutritional status and health of populations. This study estimates the waste of fresh fruits in the post-harvest chain and identifies avoidable causes along the food supply chain to extrapolate good practices for the empowerment of retailers. A semi-structured questionnaire and a checklist were used in the administrative units of Yaoundé, Cameroon, from May to June 2017. Fifty fresh fruit retailers were randomly selected. Information, including socioeconomic profile, handling practices, transport, and food wastes, was analyzed. Dominant figure in the fruit market are 34-aged women. Despite significant professional experience, none of retailers received formal training. The perceived main causes of fruit waste were failure to sell, mechanical damage during transport, and storage conditions. Inappropriate packaging materials and poor hygiene were also observed, and about 40–50% of fruits did not reach the consumers' table. Nutritional education of the general population is crucial in facing the challenge of fresh fruit waste. The analysis of critical points in the post-harvest fresh fruit chain highlights good cost-effective practices. Training and empowerment of retailers represent the main measures to decrease fruit waste, in addition to nutritional training programs for the general population recommending the daily consumption of fruits for healthy life.

**Keywords:** food waste; retailers; food transport; sustainability; resource conservation; healthy lifestyle

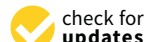

## 1. Introduction

Cameroon, as in the case of most sub-Saharan African countries, is currently facing one of its most serious food crises. According to the World Food Program (2017) [1], the number of food-insecure households in Cameroon is estimated at around 16% (about 3.9 million people), of which 1% are in a situation of severe food insecurity. There are many reasons for this. While food unavailability and inaccessibility are incriminated, food wastage is rampant. Research reports conducted by Gustavsson et al. (2011) [2] reveal that estimated annual food quantitative losses and waste in the supply chain represent approximatively 40 to 50% of the world's fresh products—30% for cereals and 20% for oilseeds, meat, and milk products. Food losses and waste refer to the decrease in quantity or quality of food along the food supply chain [3]. In the case of perishable foods such as fruits, waste also occurs along the post-harvest chain at the retail level due to a wide variety of factors. Microorganisms and residues of pesticides and other toxic chemicals or organic inputs that are used to improve crop production can also accelerate the process of post-harvest losses and lead to population health problems [4,5]. It is recognized that toxic exposures



may worsen the micronutrient status, e.g., by increasing the nutritional requirements; vice versa, imbalanced diets and micronutrients deficiencies may increase the vulnerability to the effects of toxic substances and alter body defense systems [6]. Estimates of post-harvest losses of fruits and vegetables vary considerably in developed and developing countries [7]. Low-Income countries experience the greatest post-harvest losses due to limited tools and insufficiently updated knowledge and skills on fruit production and postharvest handling practices. Environmental and climatic conditions favorable to the growth and multiplication of micro-organisms contribute to the high rate of post-harvest loss of fruits and vegetables in these countries. In some African countries, about 30% of products are lost, and this figure can rise to 50% for very perishable foods, such as fruits and vegetables [8]. Kughur et al. (2015) [9] reported 48.5% postharvest loss in Nigeria. Similarly, Zenebe et al. (2015) [10] reported 45.9% postharvest banana loss in Ethiopia, of which about 15.7%, 22.1%, and 8.1% were incurred at the farm, wholesale, and retailer levels, respectively. This rate of post-harvest losses of fruits and vegetables varies from 20 to 40% in Bangladesh [11]. In Cameroon, Kouame et al. (2013) [12] reported 10% and 20% rates of postharvest loss of amaranths and black nightshades, respectively. Most of these studies were concerned with estimating the extent of losses without, however, worries of the causes and consequences on the health of populations.

The consequences of the post-harvest losses are numerous. Post-harvest losses of fruits contribute significantly to food insecurity and affect the nutritional status and health of populations due to the qualitative and quantitative reduction of nutrients and presence of bioactive phytochemical compounds [13,14]. According to epidemiological studies, low consumption of fruits is associated with an increased risk of chronic health disorders, such as cardiovascular diseases, hypertension, hypercholesterolemia, osteoporosis, many cancers, chronic obstructive pulmonary diseases, and respiratory and mental health problems [15,16]. Considerable efforts have been made by retailers, farmers, and policy makers to reduce food loss in Cameroon, with resources devoted to planting crops, irrigation, fertilizer, and pesticides [17]. While these efforts should be renewed and supported, attention should also be paid to the avoidance of post-harvest losses. Post-harvest fruit losses represent the waste of human efforts, water, and agriculture inputs, and their reduction is less costly than increasing the production of foods [18,19]. Minimal effort can make a huge difference when applied at the right time. Unfortunately, to the best of our knowledge, no study has been yet carried out to identify the avoidable causes in post-harvest practices on which effective interventions could qualitatively and quantitatively reduce post-harvest losses of fruits and vegetables. With the purpose of implementing feasible, effective, and lasting solutions to food insecurity problems linked to post-harvest losses, this work was undertaken in Yaoundé, Cameroon, with the objectives of (i) estimating the extent of post-harvest losses of fresh fruits; (ii) identifying avoidable causes in post-harvest practices; and (iii) extrapolating good practices for feasible and effective interventions.

## 2. Materials and Methods

### 2.1. Study Area

The study was conducted from May to June 2017 (dry season) in the city of Yaoundé. This city is located in the center region of Cameroon between latitude 3°47′–3°56′ N and longitude 11°10′–11°45′ E at an altitude of 750 m covering a total area of 256 km$^2$. The annual rainfall pattern is bimodal, with an average of 1600 mm, and its average annual temperature is 24 °C. Its position in the South Cameroon plateau between forest and the Savannah expose it to a subequatorial Guinean climate with an 8-month rainy season and a 4-month dry season [20]. Yaoundé is subdivided into 7 subdivision municipalities (Yaoundé I, Yaoundé II, Yaoundé III, Yaoundé IV, Yaoundé V, Yaoundé VI, and Yaoundé VII) (Figure 1), and its population was estimated to be 1,817,524 in 2012 [20].

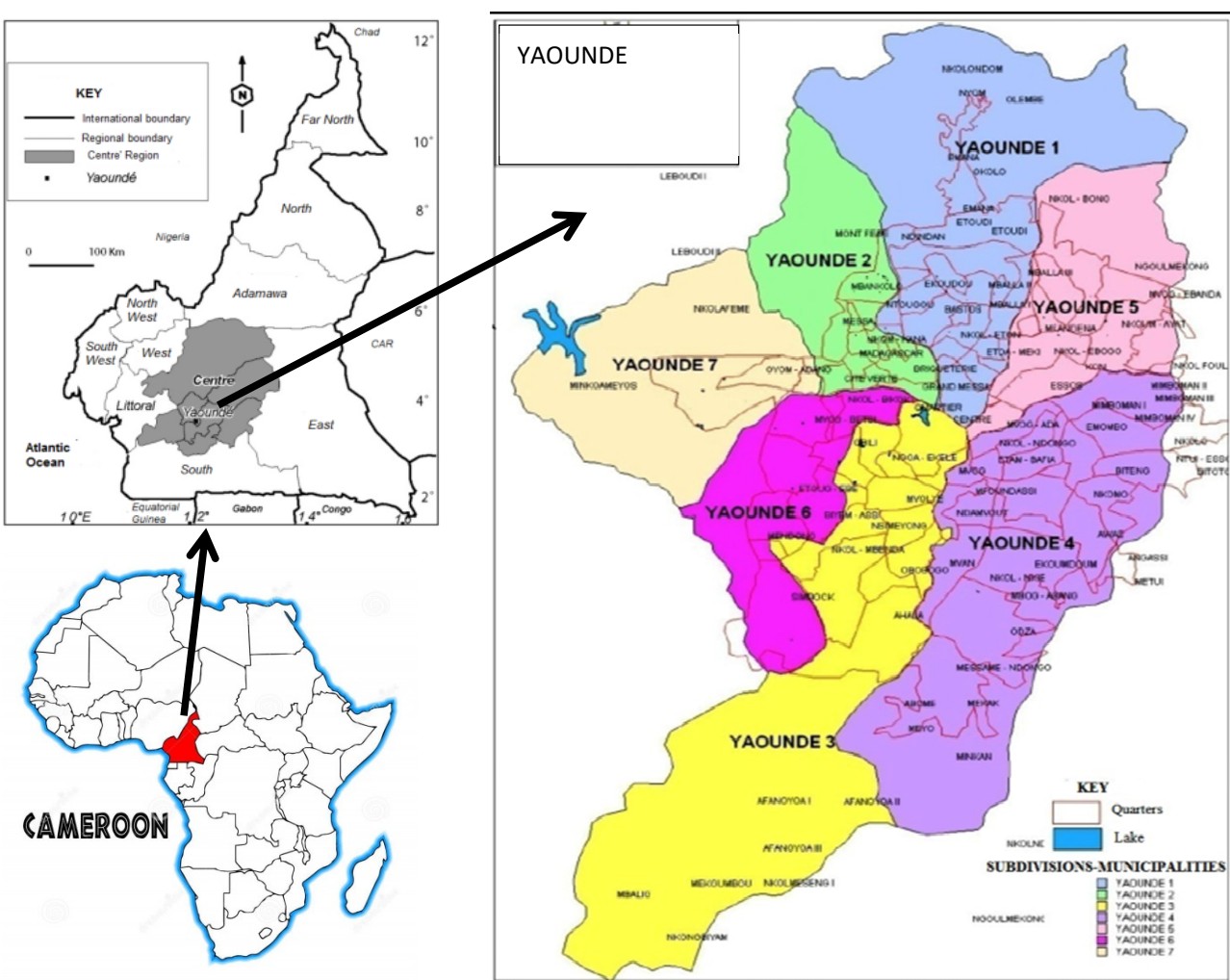

**Figure 1.** Map of Yaoundé showing the studied subdivisions municipalities. Map by «Système d"Information Géographique», Communauté Urbaine de Yaoundé, Cameroon, (2011).

### 2.2. Sampling Procedures and Data Collection

Fifty perishable fruit retailers selling at least 2 different species of fruits were randomly selected, and the "snowball" strategy was followed in the 7 administrative units of Yaoundé. The survey method was explored based on a semi-structured questionnaire and a checklist. The questionnaire was made up of closed-ended questions. It consisted of two parts: the first section focused on the retailer's socioeconomic information, while the second section regarded postharvest losses (fruit marketing practices, types of products, ways of handling, packaging and transport, estimation of food waste, and other post-harvest relevant information). Finally, the questionnaire included some open-ended questions to eventually gain salient information in the identification of problems faced by retailers and propose feasible solutions.

### 2.3. Data Analysis

The data generated from the questionnaire were analyzed using a template before applying descriptive statistics of frequency and percentage. Frequency distributions were computed and presented in tables and bar chart graphs. The data were analyzed using Statistical Package for Social Sciences (SPSS) version 20.0 Armonk, NY: IBM Corp.

### 3. Results

Table 1 presents the socioeconomic characteristics of the retailers. Over 74% of the respondents were less than 50 years old, with an average age of 34 years. Respondents (68%) had at least 5 years of experience in the fruit trade, with an average of 8–16 years of experience. Women (92%) appear to dominate the fruit market, while men account for only 8%. While 12% of the 50 respondents did not receive formal education, the vast majority of them attended primary school, and only 36% attained a level of education higher than primary school. None of the retailers surveyed received training on post-harvest fruit management.

**Table 1.** Socioeconomic characteristics of the retailers ($n = 50$).

| | Frequency | Percentage | Mean |
|---|---|---|---|
| Age (years) | | | |
| Less than 20 | 4 | 8 | |
| 21–30 | 12 | 24 | |
| 31–40 | 17 | 34 | |
| 41–50 | 9 | 18 | |
| Above 50 | 8 | 16 | |
| Mean | | | 34 |
| Gender | | | |
| Male | 4 | 8 | |
| Female | 46 | 92 | |
| Education | | | |
| No formal education | 6 | 12 | |
| Primary education | 26 | 52 | |
| Secondary education | 11 | 22 | |
| Tertiary education | 7 | 14 | |
| Experience (years) | | | |
| Less than 5 | 16 | 32 | |
| 5–10 | 18 | 36 | |
| Above 10 | 16 | 32 | |
| Mean | | | 8 |
| Training in fruit management | | | |
| yes | 0 | 0 | |
| no | 50 | 100 | |

Table 2 shows the distribution of fruit species sold by the respondents. At the time of the study, all respondents had at least two types of fruit on the counter. A total of 15 different fruit species were listed in the 7 administrative units of Yaoundé during the study period. Among those 15 species of fruits listed on the market, only 8 had a percentage greater than 50, and we considered those fruits as the most represented. These are pineapples (100%), watermelons (96%), bananas and limes (90%), oranges (76%), avocadoes (70%), lemons, (60%) and papayas (52%).

**Table 2.** Distribution of the fruit species sold by the respondents ($n = 50$).

| Types of Fruits | Frequency | Percentage |
|---|---|---|
| Pineapples | 50 | 100 |
| Watermelons | 48 | 96 |
| Bananas | 45 | 90 |
| Limes | 45 | 90 |
| Oranges | 38 | 76 |

**Table 2.** *Cont.*

| Types of Fruits | Frequency | Percentage |
|---|---|---|
| Avocadoes | 35 | 70 |
| Lemons | 30 | 60 |
| Papayas | 26 | 52 |
| Mangoes | 22 | 44 |
| Apples | 16 | 32 |
| Grapes | 16 | 32 |
| Strawberries | 10 | 20 |
| Pears | 6 | 12 |
| Guavas | 2 | 4 |
| Soursops | 2 | 4 |

The distribution based on the transportation system is presented in Figure 2. Most of the retailers (76%) use cars to transport the fruits from the main market to their shop, 24% of them use motorcycles, and 6% use pickup vans.

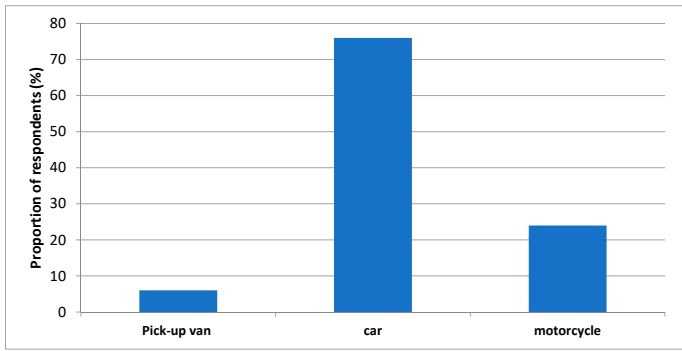

**Figure 2.** Distribution based on transportation system (*n* = 50).

According to Figure 3, most of the retailers (36%) use perforated plastic buckets to pack fruits, while 20% of them use plaited baskets and jute bags for the same operation. Twelve percent of retailers prefer cartons to pack their fruits.

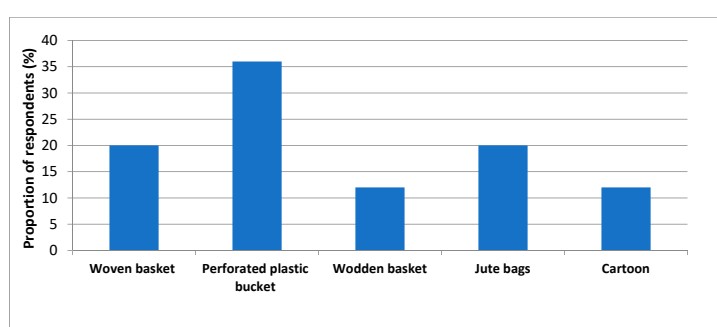

**Figure 3.** Distribution of respondents according to the types of packaging materials (*n* = 50).

Figure 4 reveals that most of the retailers (80%) receive information on fruit post-harvest practices from fellow retailers. The majority of retailers (90%) feel that they have never received information on post-harvest practices through extension agents or traditional media assets, such as television, radio, or the press. After an average shelf-life at the retailer level of 1–2 days, post-harvest losses are estimated by retailers at 20% for limes and lemon and at 50% for papayas. In addition to mangos (40%) and watermelons (40%), papayas (50%) are among the most perishable fruits according to the 50 participants in this study (Figure 5).

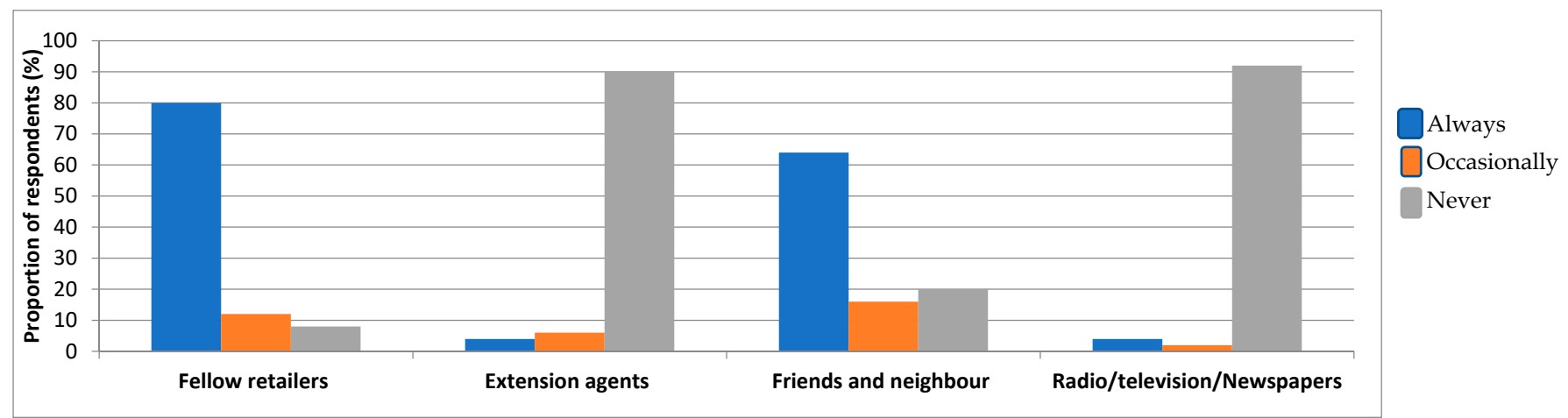

**Figure 4.** Retailers' sources of information on post-harvest practices (*n* = 50).

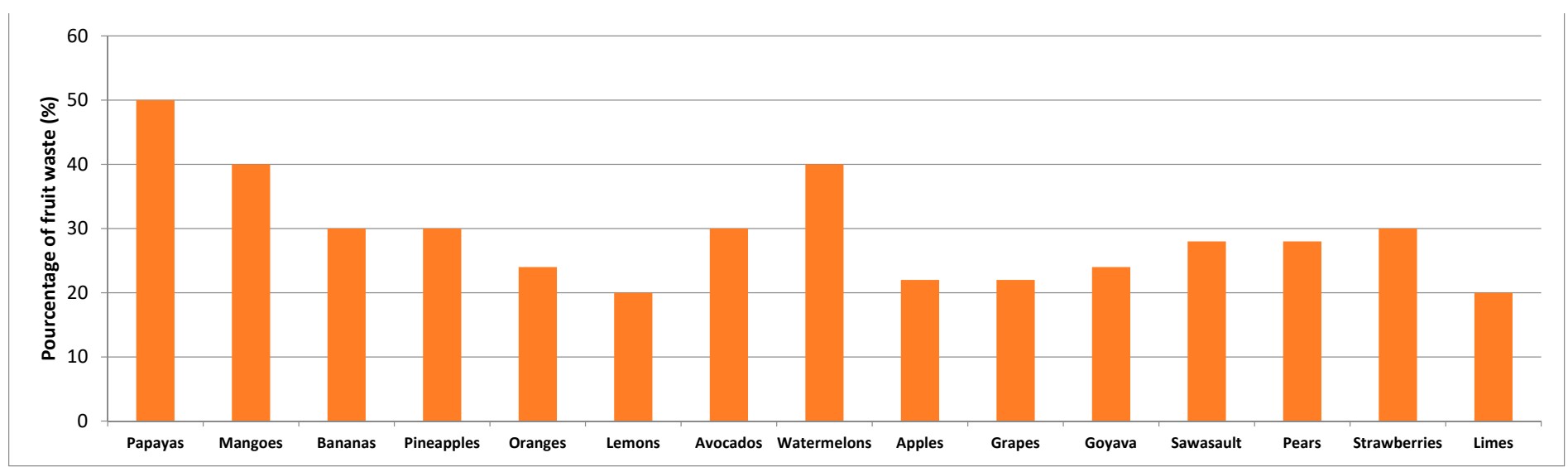

**Figure 5.** Estimation of fruit waste at the retailer level (*n* = 50).

Figure 6 shows the causes of post-harvest losses as perceived by the retailers. Most of the respondents considered that the main causes of fruit waste were failure to sell (32%), transport from the purchase point to the shops (22%), and storage conditions (18%). Other causes of post-harvest losses were rot (12%), rodent attacks (10%), insects and other organisms, poor hygiene (4%), and climatic conditions (2%). Figure 7 shows the estimated shelf life of fruit exhibited by the retailers prior to sale. For the vast majority (66%) of retailers, fruit is held for a period of 3 to 5 days before being sold to a third party, whereas 6% of participants held fruit beyond 5 days.

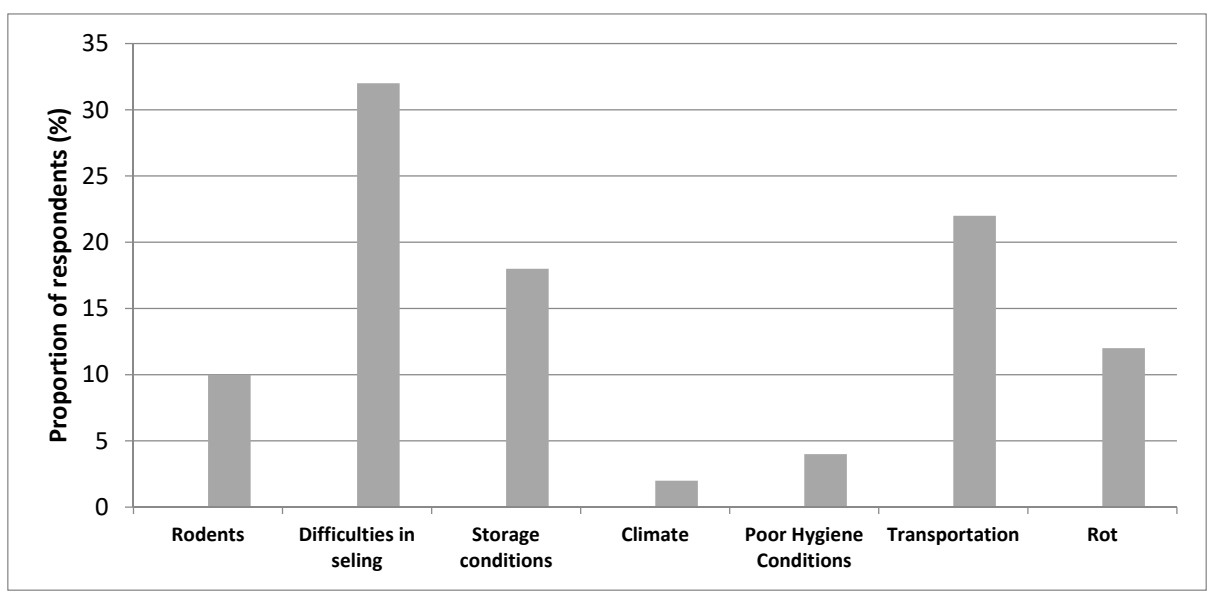

**Figure 6.** Causes of fruit waste during handling by retailers (*n* = 50).

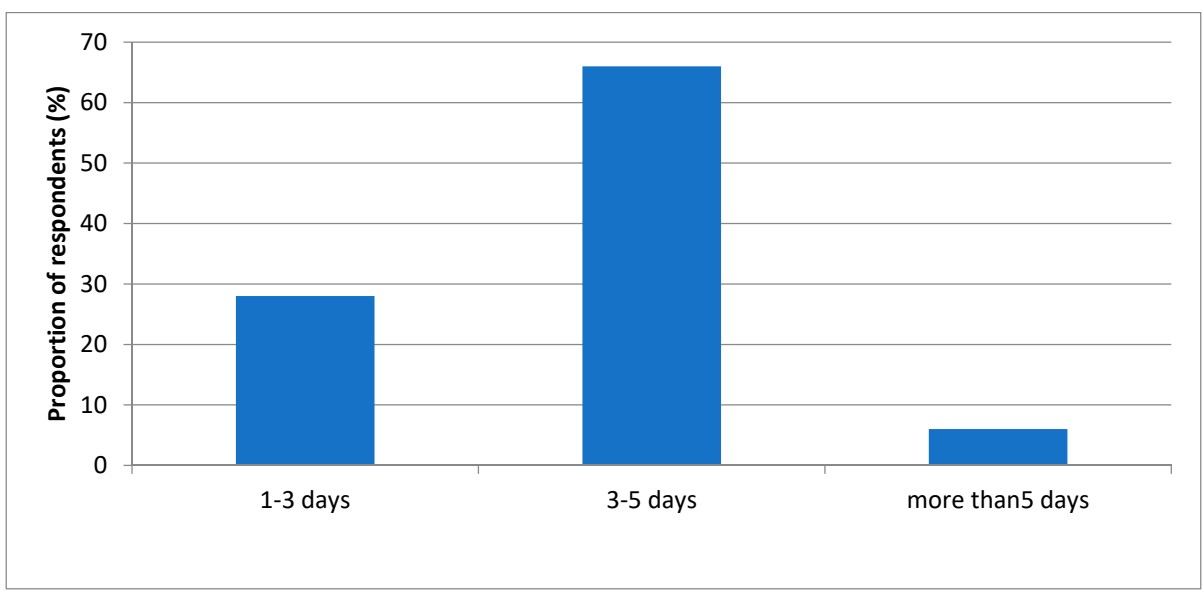

**Figure 7.** Duration of selling fruits by retailers (*n* = 50).

## 4. Discussion

The average period (8–16 years) of experience indicates that most retailers make the trade of fruits their permanent profession. The average age of retailers is the age of active, dynamic, and economically innovative youth, and also the age at which it is no longer possible to aspire for a job in the public service (the state is the main employer of young

people in Cameroon) [21]. In Cameroon, as in sub-Saharan Africa in general, women dominate the food trade activities [22,23]. The diversity of fruits sold in the Cameroonian markets (Table 2) can be explained by the diversity of climatic conditions and the availability of soils that are favorable to the cultivation of many plant species. In particular, pineapples, watermelons, bananas, and limes are available throughout the year [24]. Fruits are considered in dietary guidance because of their high concentrations of dietary fiber, vitamins, minerals, especially electrolytes, and, more recently, because of their concentration of phytochemicals, especially antioxidants [14–25]. Increasing fruit consumption is an established health-promoting behavior. Several studies have correlated low intake of fruits with chronic health disorders (hypertension, osteoporosis, cardiovascular disease, hypercholesterolemia, cancers, and respiratory and mental health problems [15–25].

The gap in education and training in the management of post-harvest fresh fruits (Table 1) has a considerable impact on the shelf-life of marketed fruits and fruit waste [26,27]. Large amounts of fruits are rejected because they do not meet the standards of shape, appearance, color, and size. These amounts of rejected fruits should also be considered as wasted nutrients; conversely, such stores of nutrients could be consumed by those without access to a healthy diet that meets their daily requirements of essential vitamins and minerals [28]. Transport can also significantly contribute to post-harvest losses through mechanical and physiological damage of the fruit. "Clandos" cars used by the majority of retailers (Figure 2) are not suitable for transporting fresh fruits because they are not well ventilated [29]. In addition, the improper loading of fruits in piles or piles of bags (Figure 8a) can damage the fruits due to the shaking of the vehicle, especially on corrugated roads [26].

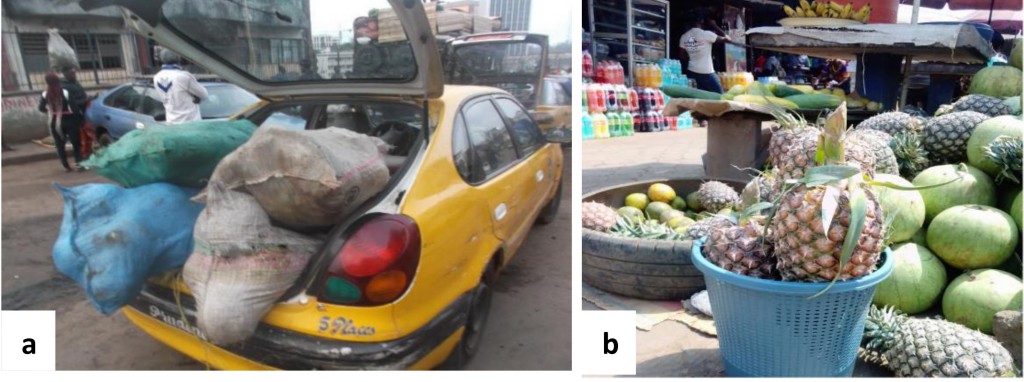

**Figure 8.** (**a**) Retailers transport watermelons in Yaoundé, (**b**) Retailers sell pineapples inside perforated plastic buckets in Yaoundé, Cameroon, 2020. Photo by Aristide Kamda. "Narrative prevention project" (www.noodlesonlus.org) © courtesy of Noodles.

The packaging materials used by retailers are cheap and widely available (Figure 3), and perforated plastic buckets are the most commonly used form (Figure 8b). However, these packaging materials have several disadvantages: the sides are sharp, they are too deep, and they both bruise the product and cause it to be jarred and/or compressed [30,31]. The use of bags is inappropriate because they do not prevent mechanical degradation during falls. In addition, the interiors of bags heat up due to congestion and metabolic reactions. These conditions result in accelerated mechanical damage and fruit attacks by microorganisms [30–32]. As previously suggested in other studies, unrecognized/overlooked real-life bad practices can be highlighted by "narrative prevention", i.e., direct observation of the field or self-reports [33]. Yahaya and Mardiyya (2019) [19] reported that in order to effectively increase the shelf-life of fruits, some cheap packaging techniques and materials such as polyethylene films, paper board boxes lined with polyethylene, and other materials may be of valuable importance and provide good protection for the fruit against dry air and microorganisms. For the same purpose, the use of appropriate chemicals substances (e.g., fungicide, gibberellic acid, and maleic hydrazide) and physical procedures (e.g.,

ionizing radiation, and in-package desiccants) during the post-harvest stage may extend the shelf-life of the product and make it available over a long period of time by protecting it from attacks by pathogens and other environmental damage [19–27]. Noticeably, the adoption of chemical solutions must undergo proper risk assessment and good practices. The use of chemicals and food contact materials must be authorized (doses, frequency, and instructions for sustainable use) based on risk assessment for the consumer, the operator, and the environment. Moreover, consumers must be educated on the importance of deep washing with clean water before consumption to protect their health and the health of their community, as well as on good practices for plastic waste collection. Eco-friendlier solutions are feasible. Plant extracts could be a useful alternative to synthetic fungicides, bactericides, and virucides in the post-harvest handling of fruits to prevent fungal, bacteria, or viral infections by pathogenic microorganisms [33,34]. Fruits passed through water emulsified with mustard oil often have an increased shelf life [35,36]. The main importance of wax coating here is to reduce evaporation and respiration [19]. Various types of fruit coating materials (e.g., polysaccharide-, protein-, lipid-, and composite-based coatings) are available for major fruits, such as bananas, mangos, pineapples and avocadoes, to extend their shelf life and slow down food decay by retarding ripening, dehydration, microbial invasion, and growth. These are reported to effectively extend the post-harvest life, minimize water loss, reduce chilling injuries and fight against post-harvest disease [37]. Jung et al. (2000) [38] developed an egg-sourced albumin coating reinforced with nanocrystalline cellulose that can be made from waste materials. When coated onto bananas, avocadoes, papayas, and strawberries, the shelf-life was extended by a week, with reduced external browning and internal ripening. The coatings are safe to ingest but are also easily removed through washing [38]. Unfortunately, the present study did not directly cover the effects of distance on fruit quality. In order to reduce post-harvest losses due to transport, fruits must be packed and stacked in well-ventilated containers. Roads should be well laid out to avoid jolts and vibrations, and transport must be carried out during cool times of the day [7,19]. Roads and distance to the place of supply can have a significant impact on fruit losses and wastes [39]. Vibrations resulting from transport transfer ripples and irregularities onto products, thereby greatly contributing to post-harvest losses [40]. Mechanical damage increases the likelihood of spoilage because pathogens enter through wounds. Transport conditions and storage were cited by Mashau et al. (2012) [41] as main causes of post-harvest fruit losses and wastes in Limpopo Province, South Africa. Poor fruit storage conditions lead to cross-contamination. Proper facilities like refrigerated storage containers for fruits are often not available or inadequate. According to Hassan et al. (2010) [42], the lack of refrigeration and cold rooms is the most influential factor in big postharvest losses experienced in low income countries.

Spoilage and physical injury (Figure 6) are common problems of fruit vendors in sub-Sahara Africa [43–45]. The acceleration of fruit decay once in the market is explained by an increase in ethylene production due to multiple traumas experienced by fruit during transport, handling, and storage [46]. This can also be explained by high temperatures that accelerate metabolism and cellular respiration [47]. Due to the very high metabolic activity, fruits have a very short shelf life, which causes high post-harvest losses. Regarding hygiene, rodent control and strict maintenance of care should be maintained at a high standard in commercial areas. Bacteria (*Salmonella* spp., *Campylobacter* spp., *Escherichia coli*, and *Listeria monocytogenes*), viruses (noroviruses), and parasites (*Angiostrongylus*, *Ascaris*, *Echinostomes*, and *Taenia solium*) attack fruits such as papayas, watermelons, mangos, and bananas because of their nutritional composition, their high moisture content, and their pH favorable to bacterial infection [48]. The action of these organisms alters the quality of fruits and leads to the loss of its economic, social, and nutritional value, with a significant impact on the environment, the nutritional status, and the health of the population [17]. Ingestion of these organisms and/or their toxins may also cause serious diseases in humans, such as diarrheal diseases (*Campylobacter* spp.), salmonellosis (*Salmonella* spp.), *Entamoeba histolytica* infection, and hepatitis (hepatitis A) [16,36]. On the other hands, lemon and limes, which

are acidic fruits, are more resistant to the action of bacteria (Figure 5). The estimated extent of post-harvest losses of perishable fruits held by retailers in Yaoundé, Cameroon, ranges from 20% to 50% during the dry season (Figure 5) after an average shelf-life at the retailer level of 1–2 days (Figure 7). During this holding time, fruits are exposed to weather (sun and rain), dust, and other environmental factors, causing changes in texture, aroma, flavor, and deterioration [49]. In general, high temperatures during the dry season have a negative impact on the shelf life of fruits, as most factors that destroy or reduce the quality of the product occur more quickly as the temperature increases [13]. During the rainy season, characterized by high rainfall, high relative humidity, and relatively high temperatures, post-harvest losses of fruits are more important. These losses are caused by the development of molds, the acceleration of decay, and the degradation of roads [50]. With all of these constraints, it is necessary for the nutrition programs set up by NGOs and public authorities to bring together all possible forms of approaches across the entire value chain that contribute to reducing the level of losses that occur during and post harvesting of fruits and all food crops. It is essential for the authorities and all stakeholders in the chain of production, conservation, and distribution to develop actions aimed at mitigating the impact of climate conditions; improve the state of roads; and reduce the level of exposure of fruits to biological, chemical, or physical contaminants.

Muhammad et al. (2012) [29] report that retailers lack general knowledge of postharvest handling despite their years of experience, and that this could be responsible for the huge waste of fruits (and vegetables) in sub-Saharan Africa. Only scant information reaches retailers from community relays, media, social networks, videos, pamphlets, and government communications. The main sources of information are fellow retailers (Figure 4) and, word of mouth, and this testifies to the solidarity among retailers, who in Cameroon, are organized in the "Association des Bayam-selam" (ASBY), Yaoundé, Cameroon, established in 2004 in spite of the insufficient training and education offered by government bodies. This is confirmed by the consistency of information from the different administrative units of the city. Finally, yet importantly, fruit waste is closely linked to consumers' competences regarding a healthy diet. Most retailers mentioned failure to sell fruits, thus confirming the poor awareness of consumers regarding the healthy value of fruit. Despite their low cost and easy access, the daily consumer demand of fresh fruits is low.

## 5. Conclusions

The post-harvest losses of fruits in the city of Yaoundé are caused by the state of the roads, failure to sell, storage conditions, poor hygiene, the transportation system, and the nature of inappropriate packaging materials of fruits. Despite it not being directly investigated in the present study, transport distance should also feature among factors possibly affecting fresh fruit quality. This study highlights how strategies to reduce food waste may include price reductions, donation practices, and consumption by sellers. Notably, the analysis of critical points in the fresh fruit chain highlights the roles of (1) good practices for retailers to improve fruit shelf-life and decrease losses, and (2) the dietary habits of the consumers. On this basis, future training and research efforts should focus on (i) the training and empowerment of retailers and their associations with good practices, and (ii) research partnerships with nutritional training programs to simultaneously support retailers, improve food and nutrition security, and promote the health of the general population.

**Author Contributions:** Conceptualization, K.S.A.G. and E.F.; funding acquisition, K.S.A.G., C.F.; and E.F.; methodology, K.S.A.G.; and E.F.; software, C.F.; validation, K.S.A.G., R.P., C.F.; and E.F.; formal analysis, K.S.A.G.; and R.P.; investigation K.S.A.G.; and R.P.; resources, K.S.A.G., R.P.; and E.F.; data curation, K.S.A.G., R.P., C.F.; and E.F.; writing—original draft preparation, K.S.A.G.; writing—review and editing, K.S.A.G., R.P., C.F.; and E.F.; visualization, K.S.A.G., R.P., C.F.; and E.F.; supervision, E.F.; project administration, E.F. All authors have read and agreed to the published version of the manuscript.

**Funding:** This research received no external funding.

**Institutional Review Board Statement:** Not applicable.

**Informed Consent Statement:** Not applicable.

**Data Availability Statement:** The data presented in this study are available on request from the corresponding author.

**Acknowledgments:** The paper stems from the Narrative Prevention project of the Pan-African NGO Noodles (Nutrition, Food Safety and Wholesomeness. Prevention, Education and Research Network, www.noodlesonlus.org). The authors acknowledge Tuscia University, Viterbo, Italy, for granting the presentation of this work at the FANUS conference, Kigali, Rwanda, August 2019. The Cameroonian association named "Association des Bayam-selam" is engaged in empowering retailers with good practices for food security, including food safety. This will facilitate synergies with programs for nutrition security, healthy diets, and a healthy environment run by Noodles.

**Conflicts of Interest:** The authors declare that there are no conflicts of interest.

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
