# Peer review of "Waste of Fresh Fruits in Yaoundé, Cameroon: Challenges for Retailers and Impacts on Consumer Health"

_agriculture, doi:10.3390/agriculture11020089_

Round 1
Reviewer 1 Report
It is advisable to double check the graphs, in particular: Fig. 2,3,5,6,7: value scales and data labels are missing;
The graph chosen for figure 2 is not easy to read;
132: lifespan or shelf life;
199: distance is an important element but previously it is never taken into consideration, why? are there data related to the case study?
The conclusions must be improved
Author Response
"Please see the attachment."

Reviewer 2 Report
The paper has to be significantly improved. It is not so well edited (specifically all the figures) and it misses of a well structured division of sections. Discussion are too much long and conclusions too much short.
More details are reported in pdf file in attachment.
Some general recommendations:
Introduction (to revise)
For example: which is the novelty of this study? There are some other similar studies carried out on this topic? if so, please, include into the introduction a short but exhaustive literature review about your research.
Material and method (to revise)
For example: Too much poor information about the questionnaire.
Results (to revise)
Specifically, the quality of figures.

Round 2
Reviewer 2 Report
The authors followed the suggestions. The paper is now ready to be published.
